# *MYC* activation and *BCL2L11* silencing by a tumour virus through the large-scale reconfiguration of enhancer-promoter hubs

C David Wood[1], Hildegonda Veenstra[1], Sarika Khasnis[1], Andrea Gunnell[1], Helen M Webb[1], Claire Shannon-Lowe[2], Simon Andrews[3], Cameron S Osborne[4], Michelle J West[1]*

[1]School of Life Sciences, University of Sussex, Brighton, United Kingdom; [2]Institute of Immunology and Immunotherapy, College of Medical and Dental Sciences, University of Birmingham, Birmingham, United Kingdom; [3]Bioinformatics Group, Babraham Institute, Cambridge, United Kingdom; [4]Department of Genetics and Molecular Medicine, King's College London School of Medicine, Guy's Hospital, London, United Kingdom

*For correspondence: m.j.west@ sussex.ac.uk

Competing interests: The authors declare that no competing interests exist.

**Abstract** Lymphomagenesis in the presence of deregulated *MYC* requires suppression of *MYC*-driven apoptosis, often through downregulation of the pro-apoptotic *BCL2L11* gene (Bim). Transcription factors (EBNAs) encoded by the lymphoma-associated Epstein-Barr virus (EBV) activate *MYC* and silence *BCL2L11*. We show that the EBNA2 transactivator activates multiple *MYC* enhancers and reconfigures the *MYC* locus to increase upstream and decrease downstream enhancer-promoter interactions. EBNA2 recruits the BRG1 ATPase of the SWI/SNF remodeller to *MYC* enhancers and BRG1 is required for enhancer-promoter interactions in EBV-infected cells. At *BCL2L11*, we identify a haematopoietic enhancer hub that is inactivated by the EBV repressors EBNA3A and EBNA3C through recruitment of the H3K27 methyltransferase EZH2. Reversal of enhancer inactivation using an EZH2 inhibitor upregulates *BCL2L11* and induces apoptosis. EBV therefore drives lymphomagenesis by hijacking long-range enhancer hubs and specific cellular co-factors. EBV-driven *MYC* enhancer activation may contribute to the genesis and localisation of *MYC*-Immunoglobulin translocation breakpoints in Burkitt's lymphoma.

## Introduction

Epstein-Barr virus (EBV) is associated with the development of numerous lymphomas including Burkitt's (BL), post-transplant, Hodgkin and certain NK and T-cell lymphomas. EBV was discovered in BL biopsies from sub-Saharan Africa (*Epstein et al., 1964*), where BL is endemic (eBL) and almost always EBV associated. BL also occurs world-wide as sporadic BL (sBL) and immunodeficiency-associated BL, where EBV positivity is approximately 20% and 60%, respectively (*Mbulaiteye et al., 2014*). Irrespective of origin or EBV status, the defining feature of BL is a chromosomal translocation involving *MYC* on chromosome 8 and an immunoglobulin (*IG*) gene. *MYC* translocations detected in BL involve either the *IG* heavy, or lambda or kappa light chain loci on chromosomes 14, 2 or 22 respectively. t(8:14) translocations occur in 85% of BL cases (*Boerma et al., 2009*). The position of the *MYC/IG* translocation breakpoint is usually far 5' of *MYC* in endemic (EBV positive) BL. In sporadic BL, breakpoints are in the first exon or intron, implicating different, but unknown, mechanisms in their generation (*Neri et al., 1988*; *Shiramizu et al., 1991*). The placement of *MYC* adjacent to

**eLife digest** The Epstein-Barr virus is a common virus that can cause mild illnesses as well as more severe diseases. The virus infects white blood cells called B cells and can drive the development of blood cancers, including Burkitt's and Hodgkin's lymphoma. In these cancers, the infected B cells multiply rapidly and continuously, free from the controls that exist in normal cells. This occurs because the Epstein-Barr virus can both switch on genes in the B cells that drive growth and turn off other genes that trigger cell death. To achieve this, the virus hijacks DNA regions called enhancers that are situated far away from the genes that they control. However, it was not clear how this hijacking process works.

Wood et al. set out to determine how the Epstein-Barr virus uses enhancers to switch on *MYC*, a gene that is a key driver of lymphoma development, and switch off *BCL2L11*, a gene that normally triggers cell death and prevents lymphoma.

Using human B cells that had been infected with the Epstein-Barr virus, Wood et al. showed that the virus completely reorganises the DNA loops that form between the *MYC* and *BCL2L11* genes and their enhancers. These loops allow the enhancers to contact their associated gene in order to activate it. Wood et al. found that the Epstein-Barr virus switches on the *MYC* gene by altering how certain enhancers contact the gene. This may explain how the virus causes particular changes to the *MYC* gene that are found in Burkitt's lymphoma.

Wood et al. also discovered new enhancers that control the activity of the *BCL2L11* gene. The Epstein-Barr virus prevents these enhancers from contacting and switching on *BCL2L11,* thus blocking cell death. This "silencing" of *BCL2L11* can be reversed by a specific drug that targets the silencing machinery used by the Epstein-Barr virus; such treatment led to the death of the infected cells.

It is now important to carry out further studies that determine how the Epstein-Barr virus hijacks enhancers to control other genes that are associated with lymphoma. This will tell us more about how the virus drives lymphoma development and will help to identify new ways of targeting Epstein-Barr virus-infected cancer cells with specific drugs.

highly active regulatory regions at these *IG* loci leads to constitutive high-level *MYC* expression and the uncontrolled proliferation of BL cells. Despite intensive study, the role of EBV in the development of BL is still unclear.

The oncogenic potential of EBV is evident from its potent transforming activity in vitro. On infection, resting B lymphocytes are growth-transformed into permanently proliferating lymphoblastoid cell-lines (LCLs). In common with other herpesviruses, EBV establishes a latent infection in infected cells. Nine viral latent proteins are expressed in EBV-immortalised LCLs; six Epstein-Barr nuclear antigens (EBNAs 1, 2, 3A, 3B, 3C and LP) and three latent membrane proteins (LMP1, 2A and 2B). EBNA2 and the EBNA3 family of distantly-related transcription factors (TF) (EBNA3A, EBNA3B and EBNA3C) play important roles in the transcriptional reprogramming of host B cells. The actions of these four EBV TFs results in the deregulation of numerous cellular genes involved in the control of B-cell growth and survival (*Zhao et al., 2011a*, *2006*; *Spender et al., 2002*; *Maier et al., 2006*; *McClellan et al., 2012*; *Hertle et al., 2009*; *White et al., 2010*). EBNA2, EBNA3A and EBNA3C are required for B-cell immortalisation and the continuous proliferation of infected cells (*Cohen et al., 1989*; *Tomkinson et al., 1993*; *Maruo et al., 2003*, *2006*; *Kempkes et al., 1995*). These TFs cannot however bind DNA directly; they control gene transcription through interactions with cellular DNA-binding proteins (e.g. RBP-Jκ and PU.1) (*Johannsen et al., 1995*; *Ling et al., 1994*; *Waltzer et al., 1994*, *1996*; *Robertson et al., 1995*; *Le Roux et al., 1994*; *Zhao et al., 1996*; *Robertson et al., 1996*). Following initial B-cell transformation in vivo, EBV-infected cells sequentially reduce the number of latent genes they express to enable progression through the B-cell differentiation pathway (*Thorley-Lawson and Babcock, 1999*). This allows entry into the memory B-cell compartment, where the virus persists. Many EBV-associated tumour cells display restricted patterns of viral latent gene expression that may reflect the differentiation state of the neoplastic precursor cell.

During B-cell transformation by EBV, EBNA2 plays a key role in upregulating numerous genes involved in driving cell proliferation, including the proto-oncogene *MYC* (*Kaiser et al., 1999*). Whether EBNA2 activation of *MYC* contributes to the genesis of the *MYC* translocation in BL cells however, is not known. EBNA2 contains an acidic activation domain and mediates gene activation by binding histone acetyl transferases and chromatin remodellers (reviewed in (*Kempkes and Ling, 2015*). EBNA3A, EBNA3B and EBNA3C individually and co-operatively activate and repress cellular gene expression (*White et al., 2010*; *McClellan et al., 2012*; *Hertle et al., 2009*). EBNA3B is dispensable for B-cell immortalisation by EBV, but appears to play a role in suppressing tumour formation in vivo, since its loss accelerates lymphoma development (*Tomkinson and Kieff, 1992*; *White et al., 2012*). The role of EBNA3A and EBNA3C as cellular gene repressors has been most extensively studied (reviewed in (*Allday et al., 2015*). These two viral TFs work cooperatively to silence key tumour suppressor gene. These include the cyclin-dependent kinase inhibitor $p16^{INK4a}$ and the pro-apoptotic Bcl-2 family binding protein Bim (*BCL2L11*) (*Skalska et al., 2010*; *Anderton et al., 2008*). EBNA3A/EBNA3C directed silencing is associated with the recruitment of polycomb repressor complex 1 and 2 (PRC1, 2) and the deposition of the histone H3 lysine 27 trimethyl mark (H3K27me3) (*McClellan et al., 2012*, *2013*; *Paschos et al., 2012*, *2009*; *Skalska et al., 2010*; *Kalchschmidt et al., 2016b*) At *BCL2L11*, PRC-mediated repression leads to longer-term silencing through the accumulation of CpG promoter methylation (*Paschos et al., 2009*).

By activating *MYC* to drive cell proliferation and counteracting *MYC*-triggered apoptosis by silencing *BCL2L11*, EBV TFs are manipulating the same pathways that are deregulated in non-viral lymphomas. In fact, lymphomagenesis only occurs in the presence of deregulated *MYC* expression when the p53-MDM2-p14$^{ARF}$ or *BCL2L11* apoptotic axes are disabled (reviewed in *Thorley-Lawson and Allday, 2008*). The mechanisms through which *MYC* and *BCL2L11* are deregulated by EBV TFs however, are not fully defined. Genome-wide analyses indicate that binding of long-range regulatory elements by EBNA2 and EBNA3 proteins plays a key role in cellular gene reprogramming (*McClellan et al., 2013*; *Zhao et al., 2011b*; *Schmidt et al., 2015*; *Jiang et al., 2014*; *Zhou et al., 2015*; *McClellan et al., 2012*). Indeed, upstream *MYC* enhancer regions bound by EBNA2 have been identified (*Zhao et al., 2011b*). *MYC* is one of the most commonly deregulated oncogenes in human cancers and the mapping of multiple cancer risk loci and regions of focal amplification to *MYC* enhancer regions provides strong evidence that inappropriate *MYC* expression can result from the perturbation of long-range control (*Yochum, 2011*; *Ahmadiyeh et al., 2010*; *Tuupanen et al., 2009*; *Pomerantz et al., 2009*; *Shi et al., 2013*; *Zhang et al., 2016*; *Herranz et al., 2014*). *BCL2L11* silencing by EBV has only been studied in the context of EBNA3A and EBNA3C binding to the gene promoter (*McClellan et al., 2013*; *Paschos et al., 2012*), but interestingly, inactivation of a murine-specific *BCL2L11* enhancer has recently been reported in B lymphoblastic leukaemia (*Wang et al., 2015*). This indicates that enhancer control of *BCL2L11* may be important in other contexts and may be a target for disruption in tumour cells.

Here we demonstrate that EBV EBNA2 upregulates *MYC* by activating specific upstream enhancers. This promotes upstream and reduces downstream enhancer-promoter interactions. These *MYC* enhancer interactions in EBV-infected cells are dependent on the chromatin remodelling function of SWI/SNF. At *BCL2L11*, we show that the EBV repressors EBNA3A and EBNA3C inactivate a newly-described enhancer-promoter hub in a manner dependent on the activity of PRC2. Lymphomagenesis by EBV therefore involves the hijack and reorganisation of large-scale enhancer hubs through the recruitment of specific cellular co-factors.

## Results

### The EBV TF EBNA2 targets tumour-associated superenhancers at *MYC*

To study the mechanism of *MYC* activation by the EBV TF EBNA2, we examined EBNA2 binding at the *MYC* locus using ChIP-sequencing data we obtained previously from two EBV-infected cell lines (*Gunnell et al., 2016*; *McClellan et al., 2013*). These included the EBV-transformed lymphoblastoid cell-line GM12878 (ENCODE Tier 1) generated by in vitro infection of resting B cells and the EBV-positive BL cell line Mutu III (*Gregory et al., 1990*). The Mutu III line is derived from a BL tumour cell line (Mutu I) which underwent broadening of its virus latent gene expression profile in culture. Both cell-lines therefore express the full panel of EBV latent genes, including EBNA2. We found that

EBNA2 bound multiple elements both upstream (−556, −428, and −186/168 kb) and downstream (+450, +570, +900 kb and +1.8 Mb) from *MYC* (*Figure 1A and B*). Upstream regions bound by EBNA2 in particular, have H3K27 acetylation (H3K27ac) signals characteristic of active enhancers (*Figure 1C*). In fact all regions bound by EBNA2, apart from the +900 kb enhancer, are classified as super-enhancers in numerous cancer cell lines and/or primary cells (dbSUPER, http://bioinfo.au.tsing-hua.edu.cn/dbsuper/ [*Khan and Zhang, 2016*]) (*Supplementary file 1*). Super-enhancers (SE) are large lineage-specific regulatory elements typically comprising multiple TF binding sites (*Whyte et al., 2013*). The −556 and −428 regions also contain susceptibility loci for chronic lympho-cytic leukaemia (CLL) and prostate cancer (rs2466024, rs2466035, rs18814048, rs16902094 and rs445114) (*Speedy et al., 2014*; *Gudmundsson et al., 2009*, *2012*). EBV therefore targets *MYC* enhancers that function in multiple cell-type specific and tumourigenic contexts.

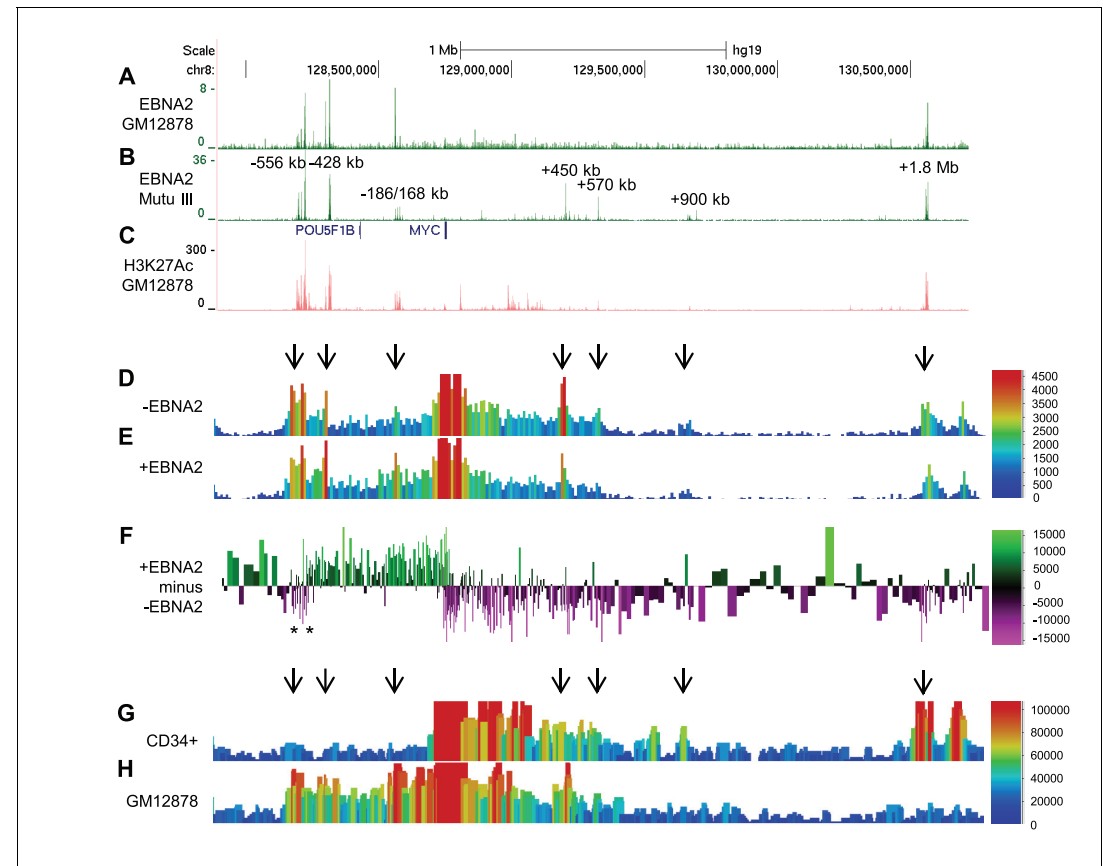

**Figure 1.** EBNA2 binding induces directional reorganisation of *MYC* promoter-enhancer interactions. (A) EBNA2 ChIP-sequencing reads in EBV-infected GM12878 cells (B) in EBV positive Mutu III BL cells (that express all latent EBV proteins). (C) H3K27ac signals in GM12878 from ENCODE. Numbering indicates the location of the major enhancer clusters relative to the *MYC* transcription start site. (D) Sequencing reads from circularised chromosome conformation capture-sequencing (4C-seq) using the *MYC* promoter as bait in ER-EB 2.5 cells expressing an EBNA2-ER fusion protein cultured in the absence of β-estradiol (-EBNA2). Reads shown are from one of two replicates. The scale bar shows reads per 10 kb window per million reads of sequencing library. (E) 4C-seq data from cells incubated in the presence of β-estradiol (+EBNA2). (F) Subtraction of -EBNA2 4C-sequence reads from +EBNA2 4C-sequence reads. The scale bar shows the normalised interaction read count difference (see Materials and methods for more details). Asterisks indicate the positions of CTCF sites. (G) Capture Hi-C sequencing reads using a *MYC* promoter bait and a CD34+ haemopoietic progenitor cell Hi-C library. Arrows denote positions where statistically significant *MYC* interactions correspond to EBNA2 binding sites. The scale bar shows reads for five merged consecutive genome fragments per million reads of sequencing library. (H) *MYC* promoter Capture Hi-C reads obtained from a GM12878 CHi-C libary.

The following figure supplement is available for figure 1:

**Figure supplement 1.** *MYC* mRNA induction in ER-EB 2.5 cells.

## EBNA2 induces large-scale directional reorganisation of *MYC* promoter-enhancer interactions

Although EBNA2 enhances the association of the −428 *MYC* SE with the *MYC* promoter (*Zhao et al., 2011b*), there is no information on how the targeting of multiple upstream and downstream long-range *MYC* enhancers by EBNA2 affects enhancer-promoter interactions across the entire *MYC* locus. We examined this using a 4C-sequencing approach.

We used a *MYC* promoter fragment as bait to capture interacting regions in an EBV-transformed LCL expressing a conditionally-active oestrogen receptor-EBNA2 fusion protein (ER-EB 2.5) (*Kempkes et al., 1995*). In these cells, EBNA2 can be reversibly inactivated through the withdrawal and re-addition of oestrogen to the culture medium, providing a useful system in which to study the effects of EBNA2 on gene transcription. Re-addition of β-estradiol to ER-EB 2.5 cells cultured for 4 days in its absence, upregulated *MYC* as previously described (*Kaiser et al., 1999*) (*Figure 1—figure supplement 1*) and resulted in the substantial directional reorganisation of interactions between EBNA2-bound regions and the *MYC* promoter (*Figure 1D–F*). In the presence of functional EBNA2, *MYC* promoter interactions with upstream elements, including the −556, −428, and −186/168 kb regions were increased. In contrast, interactions with downstream elements, including the +450 kb, +570 kb and +1.8/1.9 Mb regions were decreased (*Figure 1D–F*). Subtraction of 4C-sequencing reads obtained in the absence of functional EBNA2 from those obtained in its presence demonstrated the clear directionality of EBNA2-directed *MYC* reorganisation (*Figure 1F*). The high frequency of upstream enhancer interactions with the *MYC* promoter in EBV-infected cells was confirmed in genome-wide capture Hi-C data from GM12878 cells (*Figure 1H*). The GM12878 interaction profile contrasts with that obtained from CD34+ haemopoietic progenitor cells (*Figure 1G*) (*Mifsud et al., 2015*) and leukaemic cells (*Shi et al., 2013*), where interactions with the +1.8/1.9 Mb enhancers dominate.

We used chromosome conformation capture (3C) to verify the effects of EBNA2 on upstream *MYC* promoter-enhancer interactions (*Figure 2*). We detected 2–3-fold increases in promoter interaction frequency in the presence of EBNA2 for the largest EBNA2 binding peak in the −556 SE and for the four EBNA2 peaks in the −186/168 region (*Figure 2C and D*). EBNA2 effects on −428 SE interactions have been documented previously (*Zhao et al., 2011b*). 4C-sequencing analysis also identified a distinct region within the −556 SE that displayed reduced *MYC* promoter interactions in the presence of EBNA2 (*Figure 1F*). A 2.6-fold decrease in the frequency of interactions between this region and the *MYC* promoter was confirmed using 3C (*Figure 2C*). This region contains three binding sites for the chromatin boundary and looping factor CTCF (*Figure 2* and *Figure 2—figure supplement 1*). Two of these CTCF sites are located adjacent to one another and immediately upstream of the main −556 SE EBNA2 binding peak that shows increased promoter interactions in the presence of EBNA2. CTCF binding may therefore delineate a chromatin interaction boundary. All three CTCF sites are bound by CTCF and components of the chromatin looping complex cohesin (SMC3 and RAD21) in GM12878 cells (*Figure 2—figure supplement 1*). Using 3C, we investigated whether EBNA2 influenced interactions between these CTCF sites to change the three-dimensional arrangement of promoter interactions in this region. We found a four-fold increase in interactions between CTCF site-containing fragments in the presence of EBNA2 (*Figure 2E*). These data indicate that the process of *MYC* activation by EBNA2 through the directed reorganisation of upstream *MYC* chromatin promotes CTCF site interactions. This results in the looping out of this specific region from the enhancer-promoter chromatin hub and presumably facilitates other upstream enhancer-promoter interactions.

We conclude that EBNA2 activation of *MYC* is associated with the large-scale directional reorganisation of the *MYC* enhancer hub. EBNA2 increases upstream enhancer interactions, reduces downstream enhancer interactions and alters interactions between CTCF sites.

## *MYC* chromatin reorganisation on EBV infection differs to that induced by physiological B-cell activation

EBV infection of naïve B cells results in B-cell activation in a manner that resembles physiological B-cell activation by CD40 ligand (CD40L) and IL-4. B-cell activation by CD40L/IL-4 however results in short-term proliferation whereas EBV-infected B cells grow out into immortal cell-lines. We used 4C-sequencing to determine whether infection of resting B cells by EBV induced the same changes in

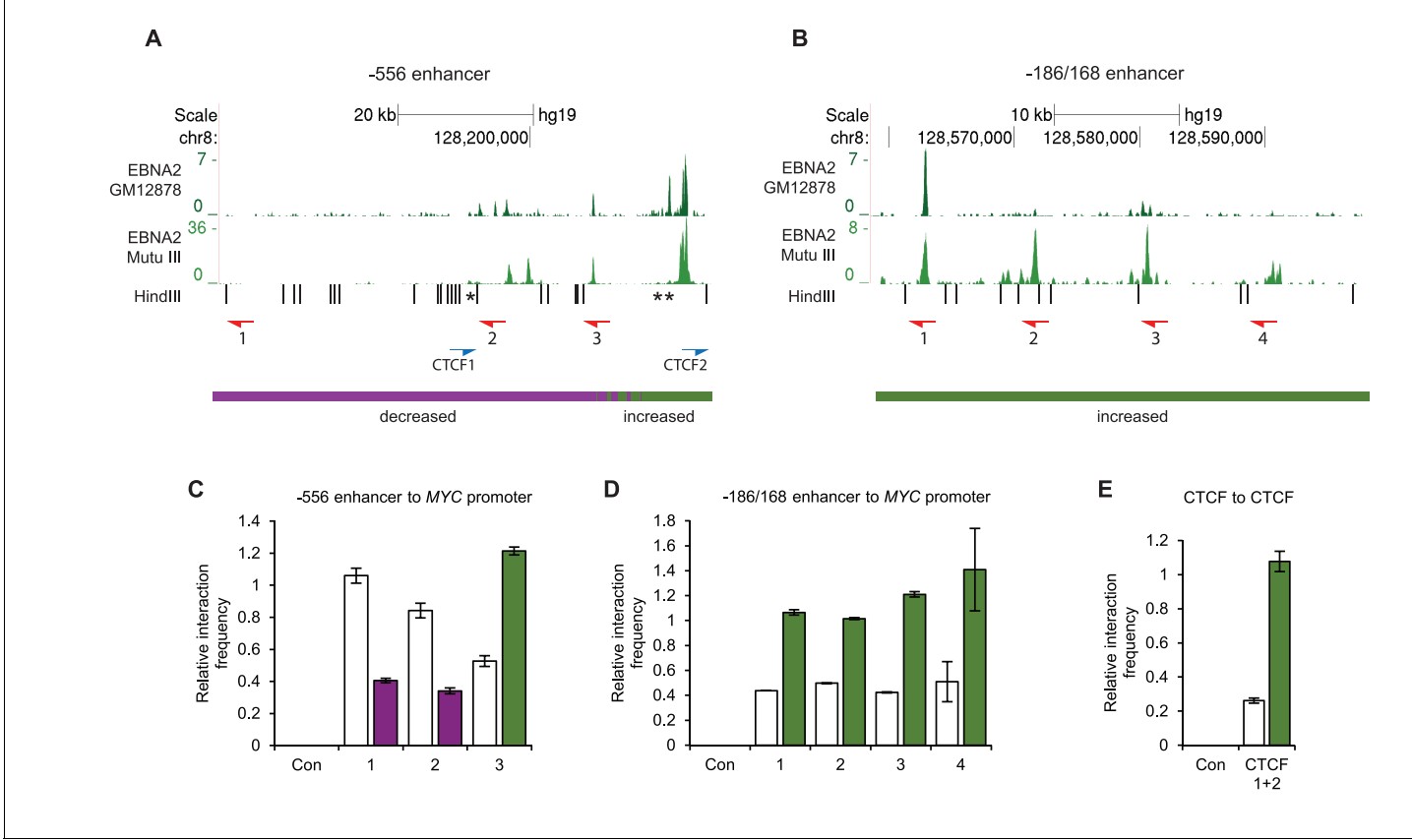

**Figure 2.** Chromosome conformation capture (3C) confirms EBNA2-induced changes at *MYC* and detects altered CTCF site interactions. (A) EBNA2 binding at the −556 super-enhancer region. The positions of the HindIII restriction enzyme sites and the primers used for 3C are indicated. Red arrows indicate the position of the *MYC* enhancer primers used for promoter interaction analysis. Blue arrows indicate the position of the CTCF site primers used to analyse CTCF site interactions. Asterisks indicate the position of CTCF sites. There are two adjacent sites at the 3' end of the region (see *Figure 2—figure supplement 1*). Primer design is unidirectional (*Naumova et al., 2012*). Purple and green lines indicate the regions that show reduced or increased interactions with the *MYC* promoter in 4C, with the transition area displaying a mix of increased and decreased interactions indicated by the checked line (see *Figure 1*). (B) EBNA2 binding and *MYC* enhancer primer positions in the −186/168 enhancer region (C) 3C analysis of interactions between the indicated −556 super-enhancer regions and the *MYC* promoter in the absence or presence of EBNA2 in ER-EB 2.5 cells. Promoter interactions with a region upstream of the −556 super-enhancer not bound by EBNA2 were analysed as a control (Con). Results show the mean ± standard deviation of signals from duplicate PCRs. (D) 3C analysis of interactions between the indicated EBNA2-bound −186/168 enhancer regions and the *MYC* promoter in the absence or presence of EBNA2. Control interaction analysis (Con) as in C. (E) 3C analysis of interactions between the CTCF sites in the −556 super-enhancer region in the absence and presence of EBNA2. CTCF site 2 interactions with the upstream control region were also analysed (Con).

The following figure supplement is available for figure 2:

**Figure supplement 1.** CTCF and Cohesin binding in the *MYC* -556 super-enhancer region.

*MYC* enhancer interactions observed in EBV-infected cell lines. We also examined whether the effects of EBV on *MYC* were distinct from any changes induced by B-cell activation by CD40L/IL-4. We found that in naïve CD19+ B cells, the *MYC* promoter interacted with the −556, −428, +450 and +570 kb enhancer regions (*Figure 3B*). This is consistent with the existence of some baseline 'static' enhancer-promoter interactions in resting cells, and the classification of the −556 region as a SE in CD19+ B cells (*Supplementary file 1*). *MYC* mRNA levels increased to maximum levels 48 hr post-EBV infection (*Figure 3—figure supplement 1*). 4C-sequencing carried out at this time point detected increases in interactions between all upstream enhancers and intervening regions and the *MYC* promoter (*Figure 3B,C and E*). This is consistent with the effects we observed on EBNA2 activation in the LCL (*Figure 1*). We also observed reduced interactions between the +450, +570 kb

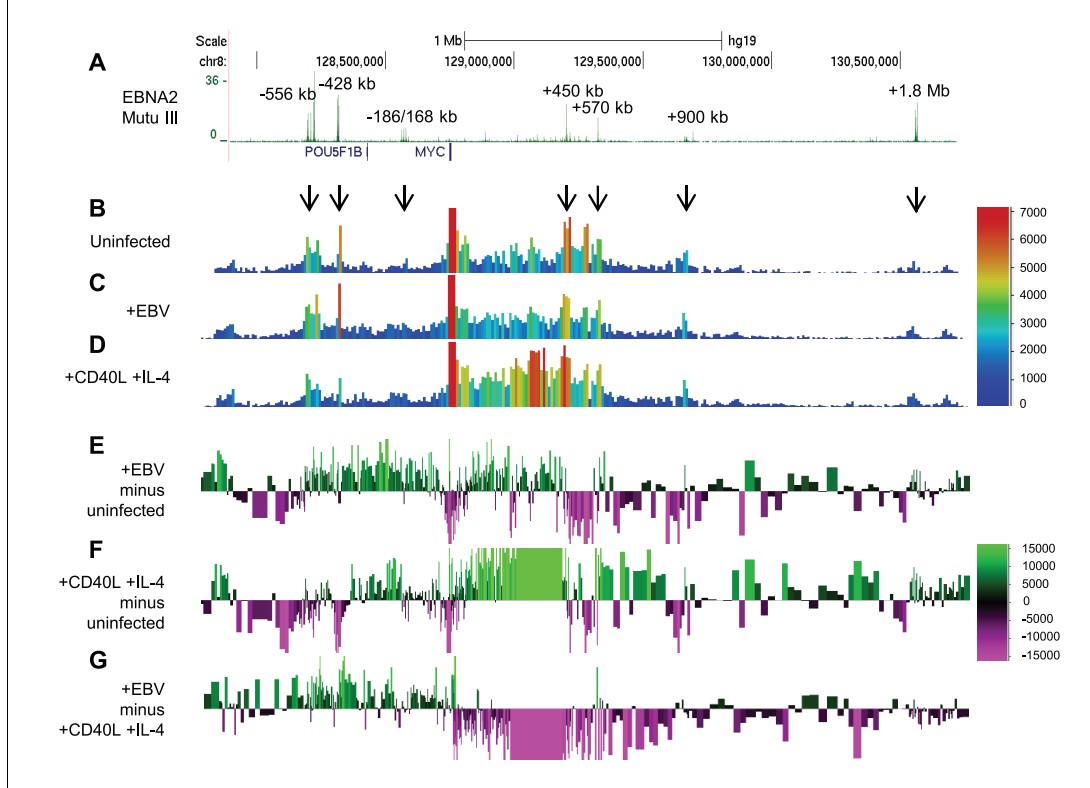

**Figure 3.** EBV infection of naïve B cells induces directional reorganisation of *MYC* promoter-enhancer interactions. (**A**) EBNA 2 ChIP-sequencing reads in Mutu III BL cells (as in *Figure 1*). Interactions captured by 4C-seq using the *MYC* promoter as bait (as in *Figure 1*) in uninfected naïve B cells (**B**), B cells 48 hr post-EBV infection (**C**) and B cells 48 hr post-stimulation with CD40L/IL-4 (**D**). Subtraction of 4C-seq reads from uninfected B cells from those obtained from EBV-infected cells (**E**) or CD40L/IL-4 treated cells (**F**). Reads shown are from both replicates combined. The scale bar shows reads per 10 kb window per million reads of sequencing library. (**G**) Subtraction of 4C-seq reads from CD40L/IL4-treated cells from those obtained from EBV-infected cells. The scale bar shows the normalised interaction read count difference.

The following figure supplement is available for figure 3:

**Figure supplement 1.** *MYC* mRNA induction on EBV infection.

and +900 kb enhancers and the *MYC* promoter on EBV infection consistent with our LCL data (*Figure 3B,C and E*). In contrast to our previous observations however, we detected some small localised increases in interactions with regions downstream from the *MYC* promoter (*Figure 3B*), which may reflect the low level of interactions present in this region in resting B cells (*Figure 3E*). In contrast, 48 hr after B-cell activation by CD40L/IL-4 we observed a reverse effect on *MYC* promoter-enhancer interactions (*Figure 3B,D and F*). Upstream interactions were reduced and downstream interactions were increased. In particular, the region +235 to +432 kb interacted with the *MYC* promoter at high level. This downstream region is not bound by EBNA2 and does not interact with the *MYC* promoter significantly in EBV-infected cells. This region does however interact at high frequency in CD34+ cells (*Figure 1G*). These data therefore highlight specificity in the remodelling of *MYC* promoter-enhancer interactions by different stimuli in B cells. This is particularly evident when the effects of EBV and CD40L/IL-4 on *MYC* promoter interactions are compared (*Figure 3G*).

In summary, EBV infection of naïve B cells reconfigures *MYC* chromatin architecture in a manner distinct from physiological B-cell stimulation. EBV infection results in the same selective enhancement of upstream enhancer-promoter interactions observed on EBNA2 activation in EBV-infected cell-lines.

## The SWI/SNF ATPase BRG1 is recruited by EBNA2 and is required for upstream enhancer-promoter interactions

We next investigated the mechanism of *MYC* activation by EBNA2. In the presence of EBNA2, we found increased levels of H3K27ac across EBNA2-bound enhancer regions, consistent with enhancer activation. EBNA2 increased H3K27ac levels seven-fold at the main *MYC* P2 promoter and 16-fold at the −556 SE (*Figure 4A*). BRG1, the ATPase subunit of the chromatin remodeller SWI/SNF, is required for the interaction of the +1.8/1.9 Mb leukaemic-cell *MYC* enhancer with the *MYC* promoter (*Shi et al., 2013*), so we examined the involvement of BRG1 in EBNA2 activation of *MYC* enhancers. We found that BRG1 associated with specific *MYC* enhancers, with highest levels at the −186/168 enhancer and the −556 SE main peak (*Figure 4B*). BRG1 also bound the +1.8 Mb enhancer but at lower levels. EBNA2 increased BRG1 binding at these sites, consistent with the ability of EBNA2 to interact with BRG1 via the Snf5 subunit of SWI/SNF (*Wu et al., 1996*). Interestingly, EBNA2 reduced BRG1 binding to the region of the −556 SE that is looped out through CTCF site association (*Figure 4B*). The effect of EBNA2 on BRG1 binding at upstream regions therefore correlates with the effect of EBNA2 on promoter interaction frequency. We next investigated whether BRG1 was required for the interaction of EBNA2-bound enhancers with the *MYC* promoter in EBV-infected cells. We found that siRNA-mediated BRG1 knockdown in GM12878 cells led to a loss of *MYC* promoter interactions with the −556, −428 and −186/168 enhancers (*Figure 4C and D*). 3C did not detect any interactions between the +1.8 Mb region and the *MYC* promoter in the presence or absence of BRG1 (data not shown), consistent with its low-level interaction frequency in EBV-infected cells (*Figures 1* and *3*). We conclude that BRG1 is required to maintain the active upstream *MYC* enhancer-promoter hub in EBV-infected B cells.

## EBV EBNA 3A and 3C silence *BCL2L11* by disrupting long-range enhancer interactions

We previously demonstrated that repression of *BCL2L11* was associated with EBNA3A and 3C-specific binding to the *BCL2L11* promoter (*McClellan et al., 2013*). However, ChIP-sequencing using an antibody that precipitates all EBNA3 proteins also revealed the presence long-range EBNA3 binding sites at the *BCL2L11* locus (*Figure 5A*) (*McClellan et al., 2012*, *2013*). These include three major sites upstream of *BCL2L11* (up to −374 kb) that lie within the neighbouring acyl-CoA oxidase-like gene *ACOXL* and three sites 310 to 587 kb downstream of *BCL2L11* (*Figure 5A*). These elements represent new putative *BCL2L11* enhancers that we designated enhancers 1–6. Enhancers 1, 4, and 6 are predicted as SEs in blood-derived primary cells or cell-lines (http://bioinfo.au.tsinghua.edu.cn/dbsuper/ [*Khan and Zhang, 2016*]) indicative of a functional role. In fact, enhancer 4 is predicted to have SE function in 30 different cell types, 22 of which are blood-derived primary or cancer cells (*Supplementary file 2*). Enhancer 4 may therefore play a key role in *BCL2L11* control in blood cells. EBNA3 binding sites are also present in the *ACOXL* promoter and the adjacent *BUB1* promoter.

ChIP-QPCR using specific antibodies demonstrated that only EBNA3A and EBNA3C bind to *BCL2L11* long-range elements (*Figure 5B–D*), consistent with *BCL2L11* silencing through the combined actions of EBNA3A and EBNA3C, but not EBNA3B (*Anderton et al., 2008*). Only EBNA3C bound at significant levels to *ACOXL* sites, with most binding at enhancers 2 and 3 (*Figure 5B–D*). Although *ACOXL* was not expressed in most B cell-lines we examined, we found that inactivation of EBNA3C in an LCL expressing a conditionally-active EBNA3C-hydroxytamoxifen fusion protein led to increased *ACOXL* expression (*Figure 5—figure supplement 1*). This points to a role for EBNA3C as a repressor of *ACOXL*.

To determine whether any of the long-range EBNA3A/3C binding sites interact with the *BCL2L11* promoter, we performed 3C analysis in a previously described BL cell-line series (BL31) (*Anderton et al., 2008*). BL31 cells derive from an EBV-negative BL and have been used to study the effects of EBV and EBNA3A, 3B and 3C on cellular gene expression by infecting them with recombinant wild-type or knock-out EBVs (*Anderton et al., 2008*; *White et al., 2010*; *McClellan et al., 2013*). The co-operative repression of *BCL2L11* by EBNA3A and EBNA3C was first described and characterised in this cell line series (*Anderton et al., 2008*; *Paschos et al., 2012*), providing an excellent background in which to study the effects of these EBV repressors on *BCL2L11* enhancer-promoter interactions. Using the *BCL2L11* promoter as bait, we found that in uninfected BL31 cells (where *BCL2L11* is expressed), the *ACOXL* promoter and all long-range elements (apart

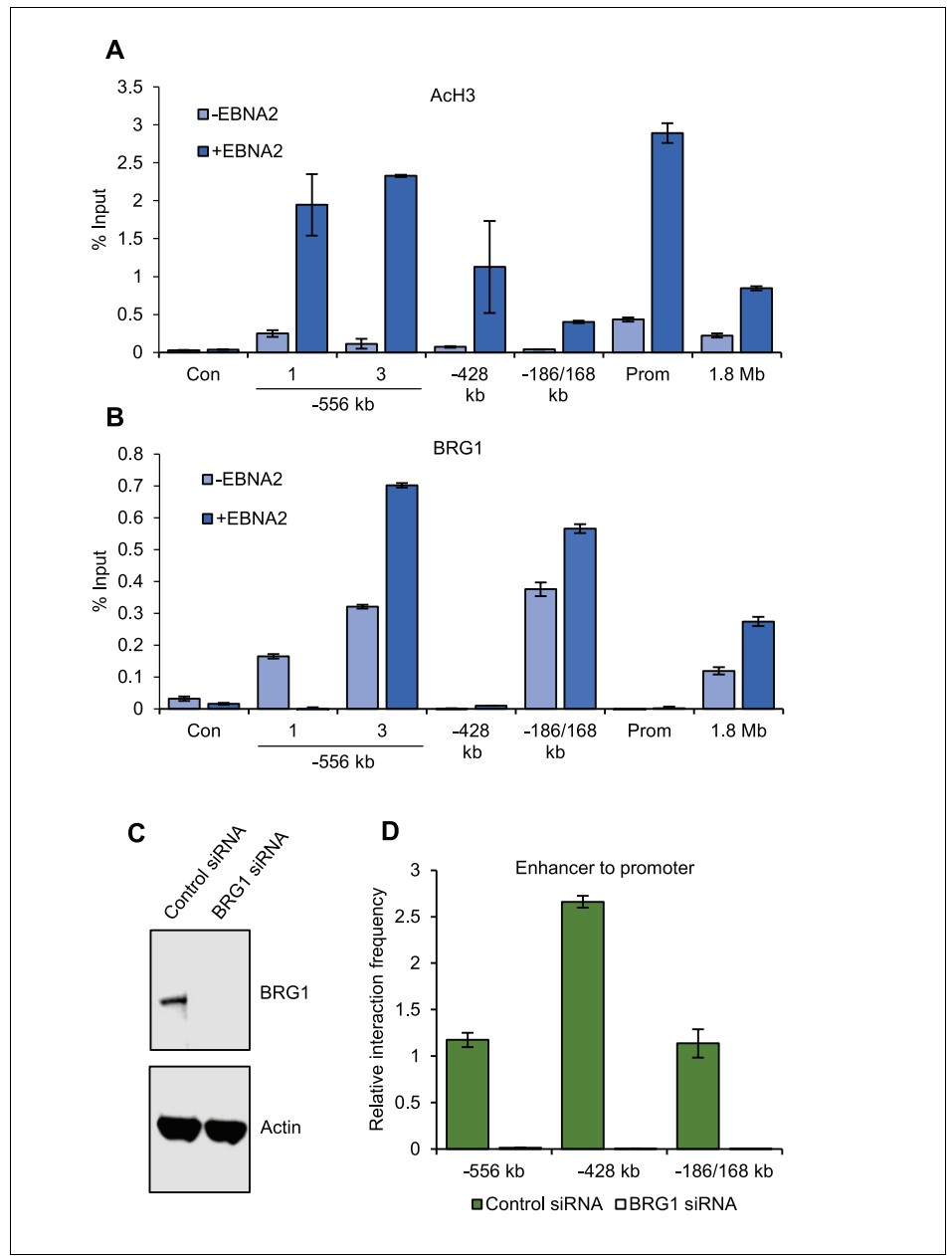

**Figure 4.** BRG1 is required for upstream *MYC* enhancer-promoter interactions in EBV-infected cells. (**A**) ChIP-QPCR analysis of H3 acetylation at *MYC* in ER-EB 2.5 cells minus or plus β-estradiol (± EBNA2). Precipitated DNA was analysed using primer sets located at the main EBNA2-bound enhancers. For the −556 SE analysis included a region where decreased interactions were observed (1) and the −556 main peak where increased interactions were observed (3) (see *Figure 2*). The signal at a control region not bound by EBNA2 (used for 3C analysis in *Figure 2*) was used as a negative control for binding (Con). Mean percentage input signals, after subtraction of no antibody controls, are shown ± standard deviation for two independent ChIP experiments. (**B**) ChIP-QPCR analysis of BRG1 binding at *MYC* in ER-EB 2.5 cells minus or plus β-estradiol (± EBNA2). (**C**) Western blot analysis of BRG1 expression in GM12878 transiently transfected with control or BRG1-specific siRNAs. Actin was used as a loading control. (**D**) Chromosome conformation capture analysis of the interaction of EBNA2-bound upstream enhancerswith the *MYC* promoter in control and BRG1 siRNA transfected GM12878 cells. Results show the mean ± standard deviation of signals from duplicate PCRs.

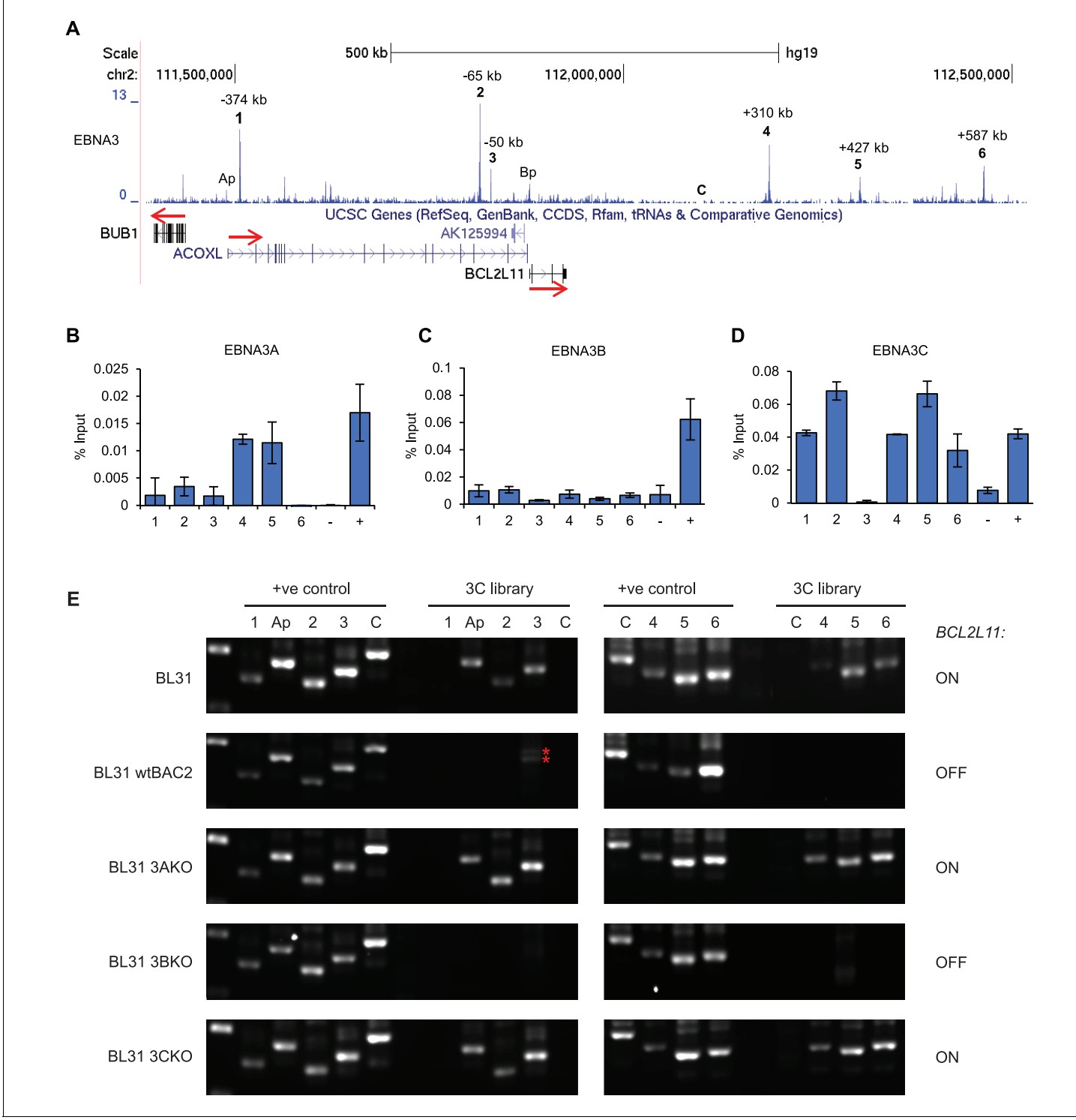

**Figure 5.** EBNA3A and 3C repress *BCL2L11* by inactivating a long-range enhancer hub. (**A**) ChIP-sequencing reads for EBNA3A/3B/3C at the *BCL2L11* locus in Mutu III BL cells. The major EBNA3-bound sites are numbered 1–6 and their location relative to the transcription start site of the *BCL2L11* promoter is indicated. Binding peaks at the *ACOXL* (Ap) and *BCL2L11* promoters (Bp) are also indicated. ChIP-QPCR analysis of EBNA3A binding (**B**) EBNA3B binding (**C**) and EBNA3C binding (**D**) in the EBV-negative BL31 BL cell-line infected with wild-type recombinant EBV (BL31 wtBAC2). Precipitated DNA was analysed using primer sets located at EBNA3A/3B/3C binding sites (binding at the *BCL2L11* promoter has been previously characterised [**McClellan et al., 2013**]). The signal at the transcription start site of *PPIA* was used as a negative control for binding (−).The previously characterised *CTBP2* binding site was used as a positive control for EBNA3A and EBNA3C binding (+). The *RUNX3* superenhancer was used as a positive control for EBNA3B binding (**Gunnell et al., 2016**). Mean percentage input signals, after subtraction of no antibody controls, are shown ±

*Figure 5 continued on next page*

*Figure 5 continued*

standard deviation for two independent ChIP experiments. (E) Chromosome conformation capture (3C) analysis of *BCL2L11* promoter interactions between enhancers 1–6 and the *ACOXL* promoter in the EBV-negative BL cell-line BL31 and in BL31 cells infected with wild-type recombinant EBV (wt BAC2), EBNA3A knock-out EBV (EBNA3AKO), EBNA3B knock-out EBV (EBNA3BKO) and EBNA3C knock-out EBV (EBNA3CKO). A control region (C) not bound by the EBNAs was also included in the analysis. Positive controls show amplification of a digested and ligated genomic PCR fragment library containing all ligation junctions. The expression status of *BCL2L11* in each line is indicated on the right. The red asterisks indicates non-specific amplification products of incorrect size.

The following figure supplements are available for figure 5:

**Figure supplement 1.** *ACOXL* is repressed by EBNA3C.

**Figure supplement 2.** Additional *BCL2L11* chromosome conformation capture controls.

from enhancer 1 within *ACOXL*) interacted with the *BCL2L11* promoter (*Figure 5E*). An active enhancer-promoter hub encompassing *ACOXL* therefore directs *BCL2L11* expression in these cells. 3C did not detect any interactions between the *BUB1* promoter and the *BCL2L11* promoter (data not shown). We detected no interactions between the *BCL2L11* promoter and four intervening control regions not bound by EBNA3A or EBNA3C, indicating that the enhancer-promoter interactions we detected were specific. (*Figure 5E* and *Figure 5—figure supplement 2*). In contrast to EBV-negative BL31 cells, in BL31 cells infected with either wild-type EBV or EBNA3B knock-out EBV (where *BCL2L11* is repressed), we observed a loss of all promoter-enhancer interactions (*Figure 5E*). Accordingly, in cells infected with EBNA3A or EBNA3C knock-out viruses, *BCL2L11* was expressed and all enhancer-promoter interactions were preserved. *BCL2L11* silencing by EBNA3A and EBNA3C is therefore associated with the inactivation of an active enhancer-promoter hub encompassing *ACOXL*.

We next investigated whether the disruption of enhancer-promoter interactions by EBNA3A and EBNA3C was also associated with enhancer chromatin inactivation. Using ChIP-QPCR we examined the binding of the PRC2 H3K27 methyltransferase EZH2 across EBNA3-bound sites. Consistent with previous observations (*Paschos et al., 2012*), we found that EZH2 was associated with the *BCL2L11* promoter in cells infected with viruses expressing EBNA3A and EBNA3C (wt BAC2 and EBNA3B KO, *Figure 6A*). We also detected EZH2 binding to all enhancers targeted by EBNA3A and EBNA3C and to the *ACOXL* promoter (*Figure 6A*). These results are consistent with enhancer inactivation either initiated by, or resulting in, PRC-associated chromatin silencing. EBNA3A and EBNA3C have been shown to induce the deposition of the PRC silencing mark H3K27me3 across the *BCL2L11* promoter (*Paschos et al., 2009*). H3K27me3 ChIP-seq data from GM12878 cells (that express EBNA3A and EBNA3C) however, also demonstrates the presence of characteristically broad domains of H3K27me3 that coincide with the locations of the EBNA3-bound *BCL2L11* enhancers (*Figure 6—figure supplement 1*). In fact a large H3K27me3 domain encompasses the entire *ACOXL* gene and the *BCL2L11* promoter (*Figure 6—figure supplement 1*). The corresponding absence of active H3K27ac marks in these regions is consistent with silencing of both genes.

To determine whether *BCL2L11* silencing by EBV could be reversed through the loss of EZH2 activity, we treated EBV-negative BL31 cells and EBV-infected BL31 wt BAC2 cells with the EZH1/2 inhibitor UNC1999. In EBV-infected BL31 cells, *BCL2L11* mRNA expression increased up to 3.6- and 5.4-fold after 8 and 18 hr treatment with UNC1999, consistent with the inhibition of PRC2-mediated gene repression (*Figure 6B and C*). In contrast, treatment of EBV-negative BL31 cells with UNC1999 resulted in only 1.8 to 2.7 fold increases in *BCL2L11* expression after 8 and 18 hr, respectively (*Figure 6B and C*). Since *BCL2L11* is not repressed by EBV in BL31 cells, these small increases may reflect the fact that even though the gene is expressed, further de-repression can be achieved through EZH1/2 inhibition. Consistent with increased *BCL2L11* expression and the pro-apoptotic function of *BCL2L11*, EBV-infected BL31 cells treated with UNC1999 also displayed large increases in Caspase 3/7 activity (*Figure 6D and E*). Caspase 3/7 activity was very low in EBV-negative BL31 cells but increased slightly in the presence of UNC1999, consistent with the smaller increases in *BCL2L11* expression. These data therefore indicate that PRC-mediated silencing of *BCL2L11* by

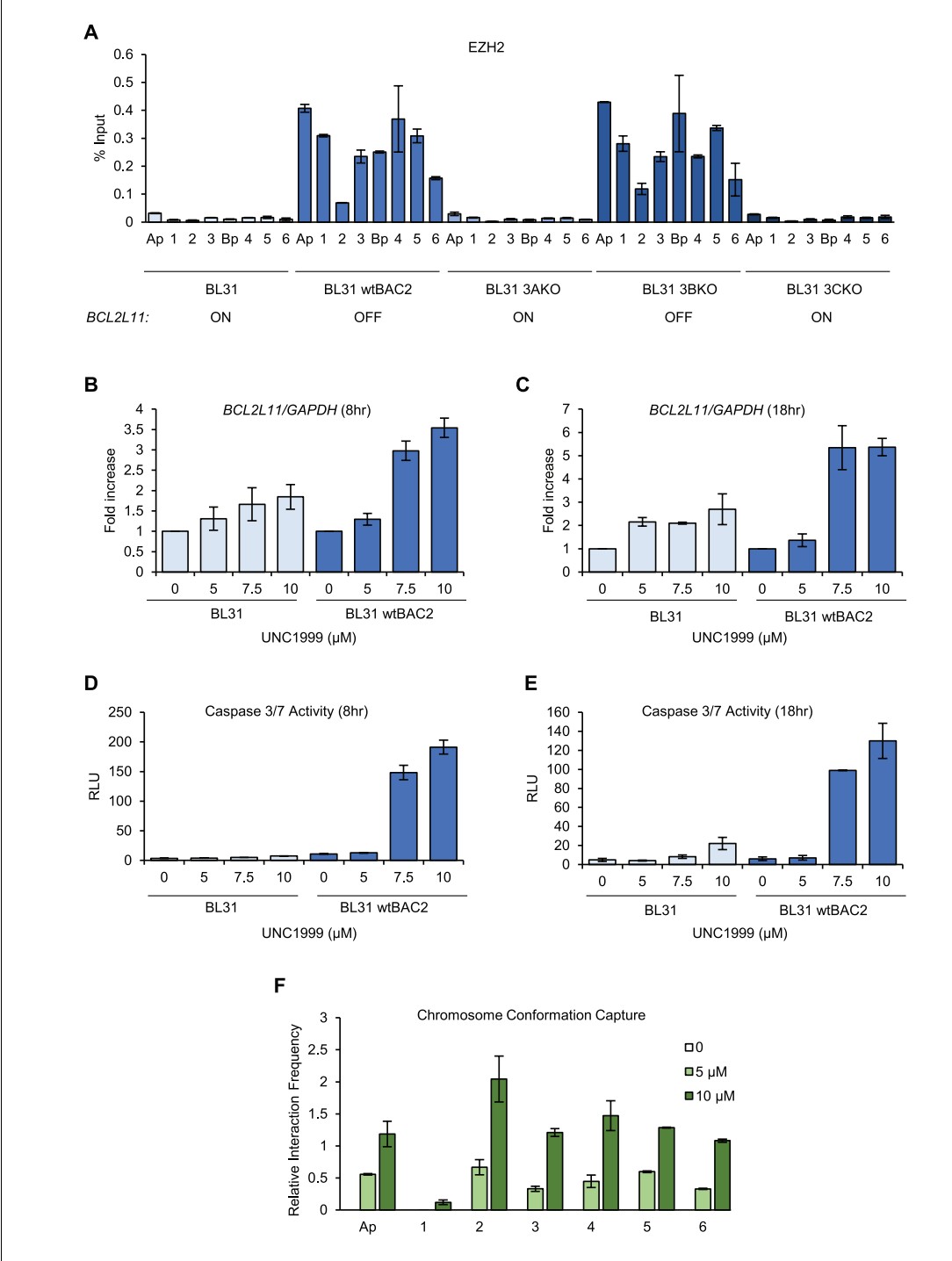

**Figure 6.** EZH1/2 activity is required for the disruption of the *BCL2L11* and *ACOXL* enhancer hub. (**A**) ChIP-QPCR analysis of EZH2 binding in the BL31 cell line series used in *Figure 5*. The expression status of *BCL2L11* in each cell line is shown. (**B**) RT-QPCR analysis of *BCL2L11* mRNA expression in EBV negative BL31 cells or BL31 cells infected with wild-type recombinant EBV (BL31 wtBAC2) treated with the EZH2 inhibitor UNC1999 for 8 hr. Signals were normalised to *GAPDH* mRNA levels and expressed as fold increase compared to untreated cells. (**C**) *BCL2L11* mRNA expression in BL31 and BL31 wtBAC2 cells treated with UNC1999 for 18 hr. (**D**) Caspase 3/7 activity in BL31 or BL31 wtBAC2 cells treated with UNC1999 for 8 hr. Caspase signals shown are corrected for the number of live cells. (**E**) Caspase 3/7 activity in cells treated for 18 hr. (**F**) Chromosome conformation capture analysis of *BCL2L11* promoter interactions between enhancers 1–6 and the *ACOXL* promoter in BL31 wtBAC2 cells treated with UNC1999 for 24 hr. Primers are as in *Figure 5* and *Figure 5—figure supplement 2*.

*Figure 6 continued on next page*

*Figure 6 continued*

The following figure supplement is available for figure 6:

**Figure supplement 1.** EBNA3A and EBNA3C-bound enhancers at the *BCL2L11/ACOXL* locus are within H3K27me3 repressed domains.

EBNA3A and EBNA3C can be reversed by EZH1/2 inhibition and results in the induction of apoptosis.

To assess the effect of EZH2 inhibition on *BCL2L11* promoter interactions, we performed 3C analysis of the *BCL2L11* locus in EBV-infected BL31 cells following UNC1999 treatment. We found that treatment with 5 or 10 μM UNC1999 led to increased interactions between the *BCL2L11* promoter and all enhancers, and between the *BCL2L11* and *ACOXL* promoters (*Figure 6F*). These data indicate that EZH1/2 activity is required for the inactivation of *BCL2L11* enhancers and *BCL2L11* silencing in EBV-infected cells.

We conclude that the increased cell survival that results from EBV EBNA3A and EBNA3C silencing of *BCL2L11* involves the recruitment of EZH2 and the inactivation of a long-range active enhancer hub encompassing the neighbouring *ACOXL* gene.

## Discussion

B-cell immortalisation by EBV plays a central role in the development of numerous B-cell lymphomas and is required for the persistence of EBV in infected hosts. EBNA2 is essential for B-cell immortalisation and the continuous growth of EBV-infected cells (*Cohen et al., 1989*; *Kempkes et al., 1995*). Upregulation of *MYC* by EBNA2 (*Kaiser et al., 1999*) plays a key role in stimulating B-cell proliferation early in infection, promoting immortalisation. Our data now show that EBV manipulates *MYC* enhancer function to drive tumourigenesis, inducing directional remodelling of enhancer-promoter interactions over 3 Mbs. EBV promotes upstream promoter-enhancer interactions and decreases downstream interactions. The *MYC* enhancer interaction landscape in EBV-infected cells is therefore distinct from leukaemia cells, where downstream enhancers are the major controllers of *MYC* transcription (*Herranz et al., 2014*; *Shi et al., 2013*).

Whether *MYC* activation by EBNA2 also plays a role in predisposing immortalised infected cells to the *MYC* translocations that characterise EBV-positive BL has not been explored. In fact, the role of EBV in the pathogenesis of BL remains an enigma, since the defining feature of BL is a *MYC/IG* translocation rather than the presence of EBV. It has been proposed that EBV may contribute to BL simply by providing a pool of cells undergoing deregulated growth, in which a genetic accident becomes more likely. Most evidence however, now points to a role for EBV in providing a survival advantage to cells that express high-levels of *MYC* by repressing *BCL2L11* (see below). Pro-survival events presumably arise through genetic and epigenetic changes induced by non-viral mechanisms in EBV-negative BLs. Given that the *BCL2L11* repressors EBNA3A and EBNA3C are expressed initially in growth-transformed B cells, but not in BL cells (which express only EBNA1), *BCL2L11* repression is likely an early event that prevents apoptosis driven by the initial activation of *MYC* by EBNA2. Since H3K27me3-mediated *BCL2L11* repression leads to CpG methylation at the *BCL2L11* promoter (*Paschos et al., 2009*) this 'hit-and-run' silencing event would provide a long-lived survival advantage to cells in which a *MYC/IG* translocation may subsequently arise. EBV increases the likelihood of a translocation event through the upregulation of AID by EBNA3C (*Kalchschmidt et al., 2016a*). The generation of double-strand DNA breaks as a result of aberrant AID activity is strongly implicated in the genesis of *MYC/IG* translocations (*Robbiani et al., 2008*; *Dorsett et al., 2007*). AID preferentially targets active enhancer regions, so our data implicate the activation of upstream *MYC* enhancers by EBNA2 in predisposing these regions to AID-induced breakpoints in EBV-infected cells.

Interestingly, *MYC-IG* breakpoints in EBV-positive eBL carrying the common t(8;14) translocation are predominantly located far upstream of *MYC* whereas breakpoints in sBL are evenly distributed between the promoter region, the first exon and the first intron (*Neri et al., 1988*; *Shiramizu et al., 1991*; *Busch et al., 2007*; *Joos et al., 1992*; *Pelicci et al., 1986*). When we examined the location of *MYC-IG* breakpoints in eBLs in light of the EBV-induced changes we detected in the chromatin

region upstream of *MYC*, we found that previous studies mapped the majority of EBV-positive eBL breakpoints to upstream of the *Eco*RI site 7 kb upstream of *MYC* (*Neri et al., 1988*; *Shiramizu et al., 1991*; *Pelicci et al., 1986*). Only eight EBV-positive eBLs have had their 5' breakpoints further mapped or sequenced (*Haluska et al., 1986*; *Neri et al., 1988*; *Joos et al., 1992*). These seven eBL cell-lines and one eBL biopsy sample have translocation junctions −215 to −46 kb upstream of *MYC* (*Haluska et al., 1986*; *Neri et al., 1988*; *Joos et al., 1992*). These upstream eBL breakpoints therefore map to the vicinity of the EBNA2-bound −186/168 enhancer. This region has not previously been studied as a long-range *MYC* control region, but has SE proper-ties in colorectal carcinoma and diffuse large B cell lymphoma cell-lines (*Supplementary file 1*). Our data therefore highlight a role for the −186/168 enhancer in the remodelling of upstream *MYC* chro-matin that may create a 'hotspot' for eBL breakpoints. In fact all upstream breakpoints (including those more than 7 kb upstream of *MYC* that have not been fully mapped) likely fall within the upstream region that displays increased promoter looping in the presence of EBV. We therefore propose that the EBNA2-directed activation and remodelling of *MYC* upstream chromatin may increase the susceptibility of this region to a translocation event initiated by the off-target activity of activation-induced cytidine deaminase (AID). In contrast to eBL, sporadic BL t(8:14) breakpoints clus-ter in two regions much further downstream that include the *MYC* promoter (−400 bp to + 150 bp) and a region immediately downstream (+420 bp to +1.2 kb) (*Busch et al., 2007*). This suggests that specific chromatin changes or other factors contribute to the localisation of *MYC* breakpoints in dif-ferent regions in the absence of EBV.

Our data also demonstrate that the mechanism of *MYC* activation by EBNA2 through upstream enhancers involves the recruitment of the SWI/SNF ATPase BRG1. Previous studies have demon-strated that AML cell growth is dependent on *MYC* activation by SWI/SNF (*Shi et al., 2013*). This dependency appears to result from a requirement for BRG1 for the interaction of the +1.8/1.9 Mb (+1.7 Mb in mouse) *MYC* enhancer with the promoter (*Shi et al., 2013*). In AML cells, BRG1 knock-down decreased downstream enhancer interactions and increased upstream enhancer interactions, suggesting some directionality in the effects of BRG1 on enhancer looping. In EBV-infected cells however, we find that BRG1 is required for the maintenance of upstream enhancer-promoter interac-tions. Our data are therefore consistent with a model where BRG1-dependent chromatin remodel-ling is required for *MYC* enhancer-promoter interactions. The specificity of BRG1 dependence however, is determined by which *MYC* enhancers are active in a particular cell-type or context.

Repression of *BCL2L11* is a key strategy employed by EBV to circumvent *MYC*-driven apoptosis and promote survival (*Thorley-Lawson and Allday, 2008*). This is consistent with observations that the loss of a single *BCL2L11* allele accelerates lymphoma development in Eμ-*MYC* transgenic mice (*Egle et al., 2004*). The tumour suppressor role of *BCL2L11* is also supported by its deletion in 40% of mantle cell lymphomas (*Tagawa et al., 2005*), silencing through CpG promoter methylation in natural killer cell lymphomas (*Küçük et al., 2015*) and its targeting by the oncogenic miR-32 and miR17-92 microRNAs (*Ventura et al., 2008*; *Koralov et al., 2008*; *Ambs et al., 2008*). We now show that *BCL2L11* repression in B cells results from the inactivation of multiple long-range enhancers. Our previous analysis focused on the low-level binding of the EBV repressors EBNA3A and EBNA3C to the *BCL2L11* promoter (*McClellan et al., 2013*). It is now clear that the *BCL2L11* promoter is controlled through a long-range regulatory hub that is inactivated by EBNA3A and EBNA3C through a process that results in the recruitment of the PRC2 methyltransferase EZH2 and is dependent on EZH1/2 activity. No direct interaction between EBNA3A and EBNA3C and compo-nents of the PRC1 or PRC2 transcriptional repressors has been reported to date. PRC-dependent and H3K27me3-associated gene silencing by EBNA3A and EBNA3C has however been convincingly demonstrated at multiple gene loci (*Skalska et al., 2010*; *Harth-Hertle et al., 2013*; *Kalchschmidt et al., 2016b*; *McClellan et al., 2012*; *2013*). The mechanism of PRC recruitment by these EBV repressors remains unclear. In fact, time-course studies have shown that the loss of active chromatin marks from gene promoters and enhancers repressed by EBNA3A and EBNA3C precedes the binding of PRCs and the deposition of H3K27me3 (*Harth-Hertle et al., 2013*; *Kalchschmidt et al., 2016b*). PRC recruitment and H3K27me3 deposition may therefore be a sec-ondary and perhaps default event.

At the *BCL2L11* locus, we have identified enhancers that control *BCL2L11* expression in human B cells, that are manipulated by EBV to repress *BCL2L11* in infected B cells. These enhancers have potential roles in *BCL2L11* control in normal and malignant cells (*Supplementary file 2*).

Interestingly, a murine *BCL2L11* enhancer located 117 kb upstream and within *ACOXL* (but distinct from the EBV-targeted enhancers described here) was recently shown to be disrupted by binding of the TRIM33 transcription cofactor to prevent *BCL2L11*-induced apoptosis in B lymphoblastic leukaemia (*Wang et al., 2015*). Importantly, we showed that EBV-induced repression of *BCL2L11* through the disruption of enhancer-promoter interactions was reversed by EZH1/2 inhibition and resulted in increased apoptosis. This provides a therapeutic rationale for the use of EZH2 inhibitors in the treatment of EBV-positive lymphomas where *BCL2L11* is repressed. Our work also revealed that the poorly characterised *ACOXL* gene is a target for repression by EBNA3C. Interestingly, *ACOXL* contains two risk loci for CLL (*Di Bernardo et al., 2008*; *Berndt et al., 2013*), one of which (rs13401811 [*Berndt et al., 2013*]) maps to a smaller EBNA3A/B/C binding peak between enhancers 1 and 2 that we did not characterise in this study. This raises the possibility that this polymorphism deregulates the function of a *BCL2L11* enhancer. *ACOXL* is also downregulated in prostate cancer tissues (*O'Hurley et al., 2015*), but further studies are required to confirm any potential role in tumourigenesis.

In summary, we show that EBV-directed lymphomagenesis involves the hijacking of long-range enhancer hubs at *MYC* and *BCL2L11* (*Figure 7*). *MYC* enhancer activation by EBV may contribute to the genesis of *MYC* translocations in BL. Enhancer-mediated control of *BCL2L11* may be exploited in other tumourigenic contexts to manipulate cell survival.

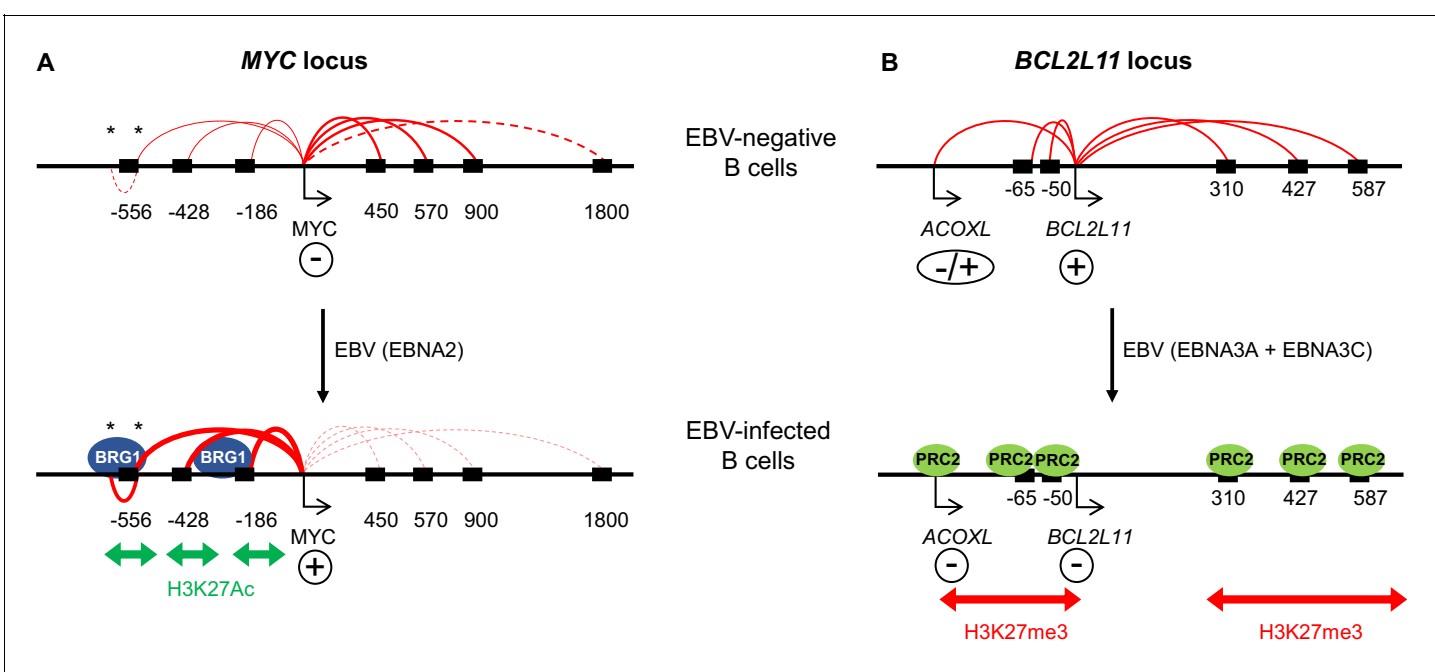

**Figure 7.** Model showing the mechanism of *MYC* activation and *BCL2L11* repression by EBV transcription factors. (**A**) In uninfected B cells, *MYC* promoter interactions with downstream enhancers dominate. *MYC* activation on EBV infection by the EBV TF EBNA2 occurs through the activation of three major clusters of upstream enhancers at −556, −428 and −186/168 kb (indicated by black boxes). This is associated with increased H3K27ac and BRG1 binding. EBNA2 promotes interactions between the *MYC* promoter and these upstream enhancers and reduces interactions with downstream enhancers. As part of this three-dimensional *MYC* enhancer reorganisation, EBNA2 also increases interactions between CTCF-bound regions (asterisks) in the −556 kb super-enhancer. (**B**) *BCL2L11* is repressed on EBV-infection by the EBV repressors EBNA3A and EBNA3C through the inactivation of multiple enhancers in regulatory hub encompassing the *ACOXL* gene. Enhancer inactivation is associated with PRC2 (EZH2) binding, increased H3K27me3 and the loss of enhancer-promoter interactions. Arrows indicate transcription start sites. Genes are indicated as expressed (+) or repressed (−). ACOXL is repressed or expressed at low-level (±).

# Materials and methods

## Cell lines

All cell lines were routinely passaged twice-weekly in RPMI-1640 media (Invitrogen, UK) containing 10% Foetal Bovine Serum, Penicillin and Streptomycin (Invitrogen). The EBV-positive latency III BL cell line Mutu III (clone 48) derives from the Mutu I latency I BL cell-line (*Gregory et al., 1990*). Mutu I cells display the 'latency I' restricted form of EBV gene expression characterised by the expression of only EBNA1. Mutu III cell clones arose spontaneously during culture of Mutu I cells and display an expanded latent gene expression pattern (latency III). The EBV-immortalised LCL GM12878 is an ENCODE Tier 1 cell line obtained from the Coriell Cell Repositories (Camden, New Jersey) (RRID: CVCL_7526). The EBV-negative BL31 cell line series infected with wild-type recombinant EBV bacmids or EBNA 3A, 3B and 3C knock-out and revertant bacmids (kindly provided by Prof M. Allday) were cultured with the appropriate selection and supplements, as previously described (*Anderton et al., 2008*). The EBV-immortalised ER-EB 2.5 LCL, expressing a conditionally-active oestrogen receptor (ER)-EBNA2 fusion protein, was provided by Prof B. Kempkes, and was cultured in the presence of β-estradiol (*Kempkes et al., 1995*). For β-estradiol withdrawal and add back experiments, ER-EB 2.5 cells were incubated in the absence of β-estradiol for 4 days, and 1 μM β-estradiol was re-added for 17 hr, prior to cell harvest. The LCL expressing conditionally-active EBNA3C is infected with recombinant EBV expressing EBNA3C fused at the C terminus to a 4-hydroxytamoxifen (HT)-sensitive murine oestrogen receptor (LCL 3CHT) and was provided by Prof M. Allday (*Skalska et al., 2010*). For 4-hydroxytamoxifen withdrawal and add back experiments LCL 3CHT cells were initially cultured in the presence of 400 nM of 4-hydroxytamoxifen (Sigma, UK) for 25 days, HT was then washed off and cells cultured for 21 days. HT was then either re-added or cells grown for a further 10 days in HT. For EZH1/2 inhibition, UNC1999 (Sigma) was added to BL31 wtBAC2 cells seeded at a density of $5 \times 10^5$ cells/ml, and cells harvested after 24 hr for mRNA and chromosome conformation capture. All cell-lines were verified as mycoplasma free.

## Isolation, infection and CD40L-mediated activation of primary resting B cells

Primary resting B cells were isolated from blood from fresh apheresis cones obtained from the NHSBT under the ethically approved study 14/WM/0001. The Peripheral blood mononuclear cells were isolated by density centrifugation on lymphoprep (Axis Shield, UK) and the B cells were subsequently purified by positive isolation using pan-CD19 Dynabeads (Dynal, ThermoFisher, UK). The Dynabeads were removed from the purified B cells by incubation with Detachabead (Dynal, ThermoFisher). Purified B cells were incubated with 2089 EBV at an MOI of 100 for 1 hr at 37°C and the unbound virus was washed off. One million uninfected purified B cells were cultured in 4 ml of medium supplemented with 50 ng/ml soluble mega CD40L (Enzo, UK) and 50 ng/ml IL4. The infected and CD40L-stimulated B cells were grown in RPMI, 10% FBS and supplemented with penicillin/streptomycin and glutamine.

## siRNA knockdown

200 nM ON-TARGETplus Human SMARCA4 (BRG1) siRNA (Dharmacon, GE Healthcare, UK; L-010431-00-0005) or ON-TARGETplus siRNA non-targeting siRNA #1 (Dharmacon, GE Healthcare; D-001810-01-05) were transfected into $5 \times 10^6$ GM12878 cells resuspended in buffer T cells using the Neon transfection and 1 pulse of 1300 V for 30 msec. Following transfection the cells were further incubated for 72 hr in normal media without antibiotics.

## Caspase 3/7 assay

BL31 and BL31 wtBAC2 cells (100 μl) were seeded into 96 well plates at a density of 20,000 cells/well and cultured for 8 or 18 hours in the presence or absence of UNC1999. An equal volume of Caspase-Glo 3/7 Assay reagent (Promega, UK) was added, and cells were incubated for 30 min at room temperature. Luminescence was measured using a Glowmax multi detection system (Promega).

## ChIP-sequencing

Previously published EBNA2 and EBNA3A/B/C Mutu III ChIP-sequencing data (*McClellan et al., 2013*) are available via GEO accession number GSE47629 and EBNA2 GM12878 data via accession number GSE76869 (*Gunnell et al., 2016*). Note that Mutu III cells have an unmapped *MYC-IG* translocation, but sequence reads are mapped to the intact *MYC* locus. EBNA2 binding sites in Mutu III cells are also detected in GM1278 cells and/or in the IB4 EBV-infected LCL (*Zhao et al., 2011b*) so the integrity of binding to these sites seems to be maintained despite their translocation.

## ChIP-QPCR

EBNA3A, EBNA3B and EBNA3C were precipitated as described previously using antibodies specific for each EBNA (*McClellan et al., 2013*). ChIP for EZH2 was carried out using 4 µg mouse monoclonal antibody (Millipore, UK; 17–662), for BRG1 using 5 µg rabbit polyclonal antibody (Santa Cruz sc-10768 (H-88 X) and for diacetylated Histone H3 using 5 µg rabbit polyclonal antibody (Millipore 06–599). Quantitative real-time PCR was carried out using specific and control primers (*Supplementary file 4*) and the standard curve method as previously described (*McClellan et al., 2013*).

## Immunoblotting

Immunoblotting was carried out as described previously (*Bark-Jones et al., 2006*; *Gunnell et al., 2016*) using anti-BRG1 (Santa Cruz biotechnology, Germany; sc-17796) and anti-actin antibodies (Sigma; A-2066).

## Chromosome conformation capture

Chromosome conformation capture assays were carried out essentially as described previously (*McClellan et al., 2013*) using *HindIII*-HF (New England Biolabs) with baits consisting of either an 11 kb fragment encompassing the *MYC* promoter, an 18.2 kb fragment encompassing the −556 kb *MYC* SE 3' CTCF site or a 10.8 kb fragment encompassing the *BCL2L11* promoter. Samples were then analysed by semi-quantitative PCR using unidirectional (rather than head to head) primers designed to amplify across ligation junctions (*Naumova et al., 2012*). Positive control PCRs across ligation junctions were carried out using libraries containing genomic DNA fragments representing expected ligation products. Positive control library fragmentswere either synthesised (Genestrings, Life Technologies) or generated from genomic PCR fragments covering the restriction sites of interest that were then digested and ligated. Titrations of positive control DNA were analysed by PCR using the same primers used for chromosome conformation capture to determine the linear range of the assay prior to analysis of the chromosome conformation capture library. Quantitation was carried out using a LiCOR imaging system following agarose gel electrophoresis and staining with GelRed (Biotium, Fremont, California). Interaction frequencies were determined by dividing the chromosome conformation capture chromatin library signal for each ligation junction product by the signal obtained for a positive control sample from within the linear range. Positive control and 3C PCRs were carried out and analysed in duplicate.

## Circularised chromosome conformation capture (4C-seq)

*MYC* promoter interacting fragments were captured using a 2.7 kb *NlaIII* fragment encompassing the *MYC* promoter, prior to further digestion by *DpnII* and 4C-seq was carried out using a previously described protocol (*Splinter et al., 2012*). Cells were passed through a 70 µm filter to obtain a single cell preparation. $1 \times 10^7$ cells were then fixed in 2% formaldehyde in the presence of 10% FCS for 10 min at room temperature. The reaction was quenched with 0.125 M glycine, and cells collected by centrifugation at 400 g for 8 min at 4°C. The pellet was resuspended in 0.5 ml lysis buffer (50 mM Tris-HCl, pH 7.5; 150 mM NaCl, 0.5 mM EDTA, 0.5% NP-40, 1% Triton X-100) with freshly added complete protease inhibitors (Roche, UK), and lysed on ice for 10 min. The nuclei were collected by centrifugation at 750 g for 5 min at 4°C, then resuspended in 0.5 ml of 1.2X CutSmart Buffer (New England Biolabs) containing 0.3% SDS and incubated for 1 hr at 37°C, while shaking at 900 rpm. Triton X-100 was then added to the nuclei to give a final concentration of 3% and the samples incubated for 1 hr at 37°C, with shaking. 200 U *NlaIII* (New England Biolabs, UK) were added to the nuclei and the samples incubated for 4 hr at 37°C, while shaking at 900 rpm. The reaction was

supplemented with a further 200 U *NlaIII* and incubated overnight at 37°C with shaking at 900 rpm. A further 200 U *NlaIII* was added, followed by an additional 4 hr incubation at 37°C while shaking at 900 rpm. The digestion reaction was stopped by incubation at 65°C for 20 min. The sample was then diluted to 7 mls with 1X ligation buffer (Roche). 50 U DNA ligase (Roche) were added to the sample, and the reaction was incubated overnight at 16°C. 300 μg of Proteinase K (Roche) were added to the sample and the reaction incubated at 65°C overnight. RNA was removed by incubation with 300 μg of RNAse for 45 min at 37°C. Following two rounds of phenol-chloroform extraction, DNA was ethanol precipitated prior to resuspension in 150 μl of 10 mM Tris-HCL pH7.5 at 37°C. The samples were then diluted to 500 μl with 1X *DpnII* buffer (New England Biolabs). 50 U *DpnII* (New England Biolabs) were added, followed by an overnight incubation at 37°C whilst shaking at 900 rpm. The digestion reaction was stopped by incubation at 65°C for 20 min. Samples were diluted in 14 mls 1X ligation buffer. 100 U DNA ligase were added and the reaction incubated overnight at 16°C. The samples were then ethanol precipitated in the presence of glycogen at −80°C, prior to purification over a QIAquick PCR purification Kit (Qiagen), and eluted in 150 μl 10 mM Tris-HCL pH7.5.

Fragments captured by the bait region were then amplified by inverse PCR using primers designed to amplify outwards from the bait region (*Supplementary file 3*). Individual forward primers included a 5' overhang of the Illumina sequence adapter P5 and a unique 'barcode' sequence and encompassed the primary *NlaIII* restriction site of the *MYC* promoter bait. Common reverse primers included a 5' overhang of the Illumina sequence adapter P7 and were designed to bind less than 100 bps from the secondary restriction site (*DpnII*) in the *MYC* promoter bait. PCR was performed using Expand Long Template Polymerase (Roche), with 3.2 μg of template and 1.12 nmol of the P5 and P7 primers for 2 min at 94°C, 10 s at 94°C, 1 min at 55°C, 3 min at 68°C for 29 cycles followed by 5 min at 68°C. 16 individual PCR reactions were carried out for each sample. The reactions were pooled and purified to separate the unused adapter primers from the PCR product using the High Pure PCR Product Purification Kit (Roche). Replicate chromatin samples were generated from the same cell batch and processed separately through each stage. Samples were combined for multiplex 100bp single-end Illumina HiSeq sequencing. Four ER-EB 2.5 LCL samples were sequenced in a single lane using barcodes TSBC01, TSBC02, TSBC10, and TSBC20. Six primary B cell samples were combined in a single lane using barcodes TSBC02, TSBC04, TSBC05, TSBC06, TSBC07, and TSBC12.

## 4C–sequencing and data analysis

Initial data extraction was performed using a custom script (available as a source code file) to strip out and separate embedded barcodes, and to remove reads from the restriction fragment immediately adjacent to the bait, where no digestion had occurred. Reads were mapped to the *Homo sapiens* GRCh37 genome assembly using bowtie 2 v2.2.7 using default parameters, and were filtered to retain only those uniquely mapping reads with MAPQ >=42. For absolute quantitation the genome was divided into 10 kb windows and quantitated with read counts normalised to the data set with the highest read coverage. For relative quantitation the genome was divided into windows each of which contained a total of 50,000 reads across all samples. Read counts for each region were then quantitated in each individual dataset, and the raw counts were corrected for the total read count to account for differing depths of sequencing. Interaction count differences were calculated by subtracting the normalised counts for one dataset from another. 4C-sequencing data are available via GEO accession GSE82150.

## Acknowledgements

We thank Prof Martin Allday and Prof Bettina Kempkes for providing cell-lines and Wouter de Laat for advice on 4C.

## Additional information

### Funding

| Funder | Grant reference number | Author |
|---|---|---|
| Bloodwise | 12035 | Michelle J West |
| Bloodwise | 15024 | Michelle J West |
| Bloodwise | 14007 | Cameron S Osborne |
| Medical Research Council | MR/J002046/1 | Claire Shannon-Lowe |

The funders had no role in study design, data collection and interpretation, or the decision to submit the work for publication.

### Author contributions

CDW, Performed experiments, Designed experiments, Analysis and interpretation of data; HV, SK, AG, Performed experiments, Analysis and interpretation of data; HMW, CS-L, Performed experiments; SA, Analysis of data; CSO, Analysis and interpretation of data; MJW, Wrote the article, Conception and design, Analysis and interpretation of data

### Author ORCIDs

Michelle J West, http://orcid.org/0000-0002-9497-9365

## Additional files

### Supplementary files

• Supplementary file 1. Super-enhancers predicted by H3K27ac signal and profile at EBNA2-bound *MYC* enhancers in normal and cancer cells by dbSUPER (http://bioinfo.au.tsinghua.edu.cn/dbsuper/[29]).

• Supplementary file 2. Super-enhancers predicted by H3K27ac signal and profile at EBNA3A or EBNA3C-bound *BCL2L11* enhancers in normal and cancer cells as in *Supplementary file 1*.

• Supplementary file 3. Primers used for 4C and 3C analysis.

• Supplementary file 4. Primers for ChIP-Q-PCR and RT-QPCR analysis.

### Major datasets

The following dataset was generated:

| Author(s) | Year | Dataset title | Dataset URL | Database, license, and accessibility information |
|---|---|---|---|---|
| C David Wood, Hildegonda Veenstra, Sarika Khasnis, Andrea Gunnell, Helen M Webb, Claire Shannon-Lowe, Simon Andrews, Cameron S Osborne, Michelle J West | 2016 | MYC activation and BCL2L11 silencing by a tumour virus through the large-scale reconfiguration of enhancer-promoter hubs | http://www.ncbi.nlm.nih.gov/geo/query/acc.cgi?acc=GSE82150 | Publicly available at the NCBI Gene Expression Omnibus (accession number: GSE82150) |

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
