## [Decision Letter]

Thank you for submitting your article "*MYC* activation and *BCL2L11* silencing by a tumour virus through the large-scale reconfiguration of enhancer-promoter hubs" for consideration by *eLife*. Your article has been reviewed by three peer reviewers, and the evaluation has been overseen by Nick Proudfoot as the Reviewing Editor and Jessica Tyler as the Senior Editor. The other reviewers have opted to remain anonymous.

Your referees have discussed their reviews with one another and the Reviewing Editor has drafted this decision to help you prepare a revised submission.

Essentially your lab's study describes the analysis of conformational changes between the promoter and multiple enhancer elements of two key regulatory genes *MYC* and *BCL2L11* following EBV viral infection (especially through expression of EBV encoded transcription factors EBNAs). Your data points to the critical role played by specific EBNAs in reconfiguring enhancer:promoter contacts resulting in *MYC* activation and *BCL2L11* repression. You also provide some interesting mechanistic data on how this occurs mechanistically. Overall all the reviewers found your study interesting but have defined several experimental and presentation weaknesses that will need to be remedied in your revised manuscript. These are listed as follows.

1) In Figure 1 it should highlighted that the -556 kb enhancer is complex. The 5' part shows reduced *MYC* promoter interaction following EBNA2 expression. Really, a blow up of this region should be shown in the main figure (as in Figure 1—figure supplement 2). The exact positions of the primers used for Figure 1 and especially Figure 1 (showing CTCF interaction) must be shown.

2) It is claimed from the data of Figure 3 that BRG1 is required for the formation of the EBNA2 induced enhancer:promoter contacts. For this they do an siRNA experiment in GM12878. It would make more sense to do the BRG1 knock-down in the +/- EBNA2 setting. This way direct or indirect effects can be distinguished.

3) While the data in Figure 4 are interesting as a discussion point, there are no new experiments here to relate the EBNA2 binding sites to these breakpoints. A direct relationship can’t therefore be concluded. Consequently we feel that Figure 4 should be removed and the translocation breakpoint connections be referred to in the Discussion.

4) The initial conformational analysis of the *MYC* locus employed 4C analysis. However the subsequent data relied just on 3C analysis (e.g. Figure 3, Figure 5 and Figure 6). We consider that these analyses should be confirmed a more rigorous 4C approach. Also 3C controls should be included such as 3C primers outside the interaction regions as negative controls as well as minus ligase and crosslinking controls. We note that the 4C-seq and differential 4C-seq figures shown are missing y-axes which makes it very difficult to estimate the magnitude of the effect. These should be added

5) In Figure 5, none of the EBNA3 proteins (A, B or C) ChIP at enhancer 3 above background. Ap and Enhancers 1 and 2 are also very weak for all three EBNA3 factors. How do the authors account for the strong ChIP signals at these enhancers with the general EBNA3 antibody (Figure 5)? EZH2 is also bound to these sites (Figure 5) and so it is not clear that EBNA3 binding is in-fact responsible for the recruitment of EZH2. These issues need consideration and comment.

6) In Figure 6 UNC1999 needs to be tested in the absence/presence of EBV infection to determine if the effects are dependent on the EBNA3-dependent recruitment of EZH2.

7) The manuscript should be thoroughly checked for clarity with a view to making the paper understandable to non experts. In particular a clearer description of the different cell types employed would be helpful. Also a final summary figure showing the predicted effects of different EBNAs on *MYC* and *BCL2L11* loci conformations and the key molecular players identified would be desirable.

---

## [Author Response]

*1) In Figure 1 it should highlighted that the -556 kb enhancer is complex. The 5' part shows reduced MYC promoter interaction following EBNA2 expression. Really, a blow up of this region should be shown in the main figure (as in Figure 1—figure supplement 2). The exact positions of the primers used for Figure 1 and especially Figure 1 (showing CTCF interaction) must be shown.*

As requested, we have included a blow up of both the -556 and -186/168 *MYC* enhancer regions to highlight their complexity and to indicate the exact positions of the primers used for 3C analysis of *MYC* enhancer and CTCF interactions. These blow ups and the 3C data now form a separate figure, Figure 2.

*2) It is claimed from the data of Figure 3 that BRG1 is required for the formation of the EBNA2 induced enhancer:promoter contacts. For this they do an siRNA experiment in GM12878. It would make more sense to do the BRG1 knock-down in the +/- EBNA2 setting. This way direct or indirect effects can be distinguished.*

Although it would be nice to perform BRG1 knock-down in the -/+ EBNA2 setting using the conditional EBNA2 cell line (EREB 2.5), this experiment is not possible due to the sensitivity of cells in which EBNA2 has been inactivated. In this cell line EBNA2 activity is controlled through β-estradiol-dependent translocation of the estrogen receptor- EBNA2 fusion protein into the nucleus. In the absence of β-estradiol, EBNA2 is sequestered in the cytoplasm and cannot activate its target genes. Because the growth of EBV-infected B cells is dependent on EBNA2, when EREB2.5 cells are cultured in the absence of β-estradiol, they stop proliferating, undergo cell-cycle arrest and 50% of cells apoptose (Kempkes et al., 1995). It would not be possible to transfect these cells with siRNA as efficient knock-down in B cell-lines requires the use of specialist systems such as the Neon transfection system (Invitrogen) that can be quite toxic to cells. We have optimised efficient BRG1 knock-down using this system in the EBV-immortalised lymphoblastoid cell-line GM12878 and these cells tolerate the procedure well. Neon transfection of sensitive cell lines however, results in substantial cell death. We appreciate that as result, we can only conclude that BRG1 is required for *MYC* enhancer-promoter interactions in EBV-infected cells and we cannot infer that this is an EBNA2-dependent requirement. We have revised the manuscript to make this point more clearly.

*3) While the data in Figure 4 are interesting as a discussion point, there are no new experiments here to relate the EBNA2 binding sites to these breakpoints. A direct relationship can’t therefore be concluded. Consequently we feel that Figure 4 should be removed and the translocation breakpoint connections be referred to in the Discussion.*

We have removed Figure 4 as requested and moved the discussion of the potential effects of EBNA2-induced *MYC* enhancer activation on *MYC* translocation breakpoints to the Discussion. We have also re-written the Abstract to move this point to the end.

*4) The initial conformational analysis of the MYC locus employed 4C analysis. However the subsequent data relied just on 3C analysis (e.g. Figure 3, Figure 5 and Figure 6). We consider that these analyses should be confirmed a more rigorous 4C approach. Also 3C controls should be included such as 3C primers outside the interaction regions as negative controls as well as minus ligase and crosslinking controls. We note that the 4C-seq and differential 4C-seq figures shown are missing y-axes which makes it very difficult to estimate the magnitude of the effect. These should be added*

We employed 4C to study the effects of EBV on *MYC* promoter-enhancer interactions because of the large number of enhancer binding sites bound by EBNA2 and the very large region involved (3 Mbs). Once key enhancer-promoter interactions had been defined using 4C, we used a 3C approach to confirm the effects of EBNA2 on these interactions. We already included a control region in this 3C analysis (now Figure 2). We then used 3C to look at the effects of BRG1 knock-down on specific *MYC* interactions (former Figure 3, now Figure 4). It is a common and acceptable approach to use 3C (and not more 4C) for follow-up experiments, where the effects of specific factors on specific interactions are investigated. We have now included an additional upstream enhancer to increase the scope of this 3C follow-up analysis.

For our analysis of the *BCL2L11* locus we used an entirely 3C-based approach because the location of the three gene promoters and EBNA binding sites at this multi-gene locus was not compatible with 4C strategies we explored using variety of different primary and secondary restriction enzymes. We were however able to delineate the promoters and EBNA3-bound enhancers using 3C, and this enabled us to study the role of individual EBNA 3 proteins using a panel of wild-type and knock-out EBV-infected cell-lines. Examining this number of cell-lines by 4C would not have been cost effective. As requested, we have included additional controls for this 3C analysis. These include three additional controls outside of the interaction regions (the original analysis already included one control region) and a no ligase experiment (Figure 5—figure supplement 2). These additional controls support the specificity of the *BCL2L11* enhancer-promoter and *ACOXL* promoter interactions that we have described.

We have added y axis scale bars to all of the 4C data as requested and fully define these scales in the figure legends, so the magnitude of the effects is clearer.

*5) In Figure 5, none of the EBNA3 proteins (A, B or C) ChIP at enhancer 3 above background. Ap and Enhancers 1 and 2 are also very weak for all three EBNA3 factors. How do the authors account for the strong ChIP signals at these enhancers with the general EBNA3 antibody (Figure 5) ? EZH2 is also bound to these sites (Figure 5) and so it is not clear that EBNA3 binding is in-fact responsible for the recruitment of EZH2. These issues need consideration and comment.*

ChIP-QPCR sometimes gives slightly different binding profiles to ChIP-seq, but we realise that the key experiment here, rather than looking in the EBV-infected Mutu III BL cell line, is to examine EBNA3 binding in the BL31 cell line series that we used for enhancer analysis. This way, EBNA3 binding can be properly correlated with *BCL2L11* enhancer-promoter interactions and EZH2 levels. We have now carried out ChIP-QPCR analysis in BL31 cells infected with wild-type EBV and confirmed the lack of EBNA3B binding at the *BCL2L11* locus. We also found that EBNA3C bound enhancers 1, 2, 4, 5 and 6 and EBNA3A bound at enhancers 4 and 5 (Figure 5). These data therefore correlate broadly with the Mutu III cell ChIP-seq analysis, where large peaks of binding were seen at enhancers 1 and 2. Little EBNA3A or EBNA 3C binding was detectable by ChIP-QPCR at enhancer 3 and ChIP-seq in Mutu III cells with a pan-EBNA3 antibody detected only a small peak at this site. These new data show that EBNA3 binding does correlate with the increased EZH2 levels detected in the same cell line (now Figure 6), although levels of EZH2 are lower at enhancer 2 than at other enhancers.

We would like to clarify however that we did not intend to imply that EBNA3A or EBNA3C recruit EZH2 through a direct interaction, so the binding profiles of these factors may not be expected to exactly correlate. Rather, our data show that EBNA3A and 3C-mediated repression of *BCL2L11* is mediated by enhancer inactivation that likely results from EZH2 recruitment (and increased H3K27me3). There is no evidence that EBNA3A and 3C directly bind to EZH2 or indeed other components of the PRC1 or PRC2 complexes, but there is good evidence that EBNA3A and 3C silencing results in recruitment of PRC complexes and H3K27me3 deposition and is dependent on PRC complex activity. How EBNA3 proteins drive the recruitment of PRC complexes to target genes however is still unclear. There is increasing evidence that EBNA3A and EBNA3C gene silencing may be initiated by the loss of active promoter marks (H3K9ac, H3K27ac and H3K4me3) and that PRC complex recruitment and H3K27me3 may occur subsequently. We have now made this point clearer and added extra discussion of this in the manuscript.

*6) In Figure 6 UNC1999 needs to be tested in the absence/presence of EBV infection to determine if the effects are dependent on the EBNA3-dependent recruitment of EZH2.*

As requested, we have now repeated the EZH2 inhibition experiments using UNC1999 in EBV-negative BL31 cells alongside the BL31 cell line infected with EBV (BL31 wtBAC2) (Figure 6). These new data show that UNC1999 treatment results in only minor increases in *BCL2L11* expression in uninfected cells compared to the large increases observed in infected cells (where the gene is repressed). The corresponding caspase assays demonstrate that there are large increases in apoptosis in EBV infected cells as a result of UNC1999 treatment that correlate with the increased *BCL2L11* levels. Caspase activity is very low in uninfected cells with only minor increases observed at high UNC1999 concentrations consistent with the small effects on *BCL2L11* expression. These data therefore support our conclusion that EBV-directed *BCL2L11* repression is dependent on EZH2.

*7) The manuscript should be thoroughly checked for clarity with a view to making the paper understandable to non experts. In particular a clearer description of the different cell types employed would be helpful. Also a final summary figure showing the predicted effects of different EBNAs on MYC and BCL2L11 loci conformations and the key molecular players identified would be desirable.*

We have re-written the manuscript to make it clearer to the non-expert by adding extra background information in the Introduction and adding extra explanations of the cell systems used. We have added a final summary figure as requested (Figure 7).